

# Transcriptome analysis of colored calla lily (*Zantedeschia rehmannii* Engl.) by Illumina sequencing: *de novo* assembly, annotation and EST-SSR marker development

Zunzheng Wei[1,2,*], Zhenzhen Sun[3,*], Binbin Cui[4], Qixiang Zhang[1], Min Xiong[2], Xian Wang[2] and Di Zhou[2]

[1] Beijing Key Laboratory of Ornamental Plants Germplasm Innovation & Molecular Breeding, National Engineering Research Center for Floriculture and College of Landscape Architecture, Beijing Forestry University, Beijing, China

[2] Key Laboratory of Biology and Genetic Improvement of Horticultural Crops (North China), Ministry of Agriculture, Key Laboratory of Urban Agriculture (North), Ministry of Agriculture, Beijing Vegetable Research Center, Beijing Academy of Agriculture and Forestry Sciences, Beijing, China

[3] Beijing Key Laboratory of Separation and Analysis in Biomedicine and Pharmaceuticals, Beijing Institute of Technology, Beijing, China

[4] Department of Biology and Chemistry, Baoding University, Baoding, Hebei, China

[*] These authors contributed equally to this work.

Corresponding author
Di Zhou, zhoudibvrc@163.com

## ABSTRACT

Colored calla lily is the short name for the species or hybrids in section *Aestivae* of genus *Zantedeschia*. It is currently one of the most popular flower plants in the world due to its beautiful flower spathe and long postharvest life. However, little genomic information and few molecular markers are available for its genetic improvement. Here, *de novo* transcriptome sequencing was performed to produce large transcript sequences for *Z. rehmannii* cv. 'Rehmannii' using an Illumina HiSeq 2000 instrument. More than 59.9 million cDNA sequence reads were obtained and assembled into 39,298 unigenes with an average length of 1,038 bp. Among these, 21,077 unigenes showed significant similarity to protein sequences in the non-redundant protein database (Nr) and in the Swiss-Prot, Gene Ontology (GO), Cluster of Orthologous Group (COG) and Kyoto Encyclopedia of Genes and Genomes (KEGG) databases. Moreover, a total of 117 unique transcripts were then defined that might regulate the flower spathe development of colored calla lily. Additionally, 9,933 simple sequence repeats (SSRs) and 7,162 single nucleotide polymorphisms (SNPs) were identified as putative molecular markers. High-quality primers for 200 SSR loci were designed and selected, of which 58 amplified reproducible amplicons were polymorphic among 21 accessions of colored calla lily. The sequence information and molecular markers in the present study will provide valuable resources for genetic diversity analysis, germplasm characterization and marker-assisted selection in the genus *Zantedeschia*.

## INTRODUCTION

*Zantedeschia* species, commonly known as calla lily or arum lily, belong to the genus *Zantedeschia* in the family Araceae. These plants are native to central and southern Africa and generally grow in marshy places, on grassy slopes, and even at forest margins. The genus *Zantedeschia* is divided into two distinct sections (*Letty, 1973*; *Singh, 1996*): section *Zantedeschia*, with one evergreen species (*Z. aethiopica* Spreng.), and section *Aestivae*, with six deciduous species (*Z. rehmannii* Engl., *Z. jucunda* Letty., *Z. elliottiana* Engl., *Z. pentlandii* Wittm., *Z. valida* Singh. and *Z. albomaculata* Baill.). *Z. aethiopica*, also called white calla lily, is characterized by a pure white spathe and leaves that do not die down in the winter. In contrast, the species or hybrids of section *Aestivae*, also called colored calla lily, are characterized by a variety of spathe colors and have leaves that die down in the winter. In addition, a new species, *Z. odorata* Perry., that is dormant in the summer and has a white spathe that can produce an invariably delicate freesia-like scent, was classified into section *Zantedeschia*. There are a number of post-fertilization incompatibility barriers between the species in section *Zantedeschia* and section *Aestivae*, including endosperm degeneration, abnormal embryo development and arrested plastid development (*Yao, Cohen & Rowl, 1994*; *Yao, Cohen & Rowland, 1995*).

As a famous flower plant, colored calla lily is very popular all over the world because of its beautiful flower spathe and long postharvest life. It is an important export flower crop for New Zealand, the Netherlands and the United States (*Tjia, 1985*; *Funnell, 1993*; *Funnell & MacKay, 1999*). The extensive commercial production of colored calla lily for cut flowers and/or planting material occurs in Auckland and Palmerston North (New Zealand), South Holland (the Netherlands), California and Colombia (the United States) and so on. The potential application of colored calla lily for commercial tuber production, cut flowers, pot flowers and even as landscape specimens is also unlimited worldwide (*Tjia, 1985*). Significant advances have been made, especially for colored calla lily, through traditional breeding practices in the floriculture industry. A multitude of cultivars and hybrids showing great variation in flower color from dark red, pink, orange, yellow to even white are available at present. In fact, most of the cultivars that are hybridized, commercially developed, and used for cut or potted ornamental purposes belong to the species *Z. rehmannii*, *Z. elliottiana*, *Z. pentlandii* and *Z. albomaculata* (*Funnell & Zantedeschia, 1993*; *Funnell & MacKay, 1999*).

Molecular markers are powerful genetic tools for gene mapping and molecular marker-assisted selection (MAS) in the breeding of several crop species (*Varshney et al., 2007*). Previous studies on colored calla lily also have reported that several dominant markers, including random amplified polymorphic DNA (RAPD) and inter-simple sequence repeats (ISSR), were used for cultivar identification and to assess genetic diversity (*Hamada & Hagimori, 1996*; *Lu et al., 2012*; *Zhang et al., 2009*; *Chen et al., 2013*; *Lu et al., 2014*). However, the current conventional breeding of calla lily in section *Aestivae* is still performed without the aid of molecular markers. It is perhaps the dominant inheritance pattern of these markers that hinders the detection of allelic information. In contrast, co-dominant markers, such as simple sequence repeats (SSRs) and single nucleotide polymorphisms (SNPs), are

the favored markers for the detection of allelic diversity. Among these markers, SSR markers have gained attention due to their multi-allelic nature, abundance throughout the genome, high reproducibility and polymorphism, adaptability to automation and high-throughput genotyping (*Morgante, Hanafey & Powell, 2002*). Nevertheless, progress in developing SSR markers in section *Aestivae* and the genus *Zantedeschia* is limited compared to that in other crop plants. Only one previous effort has produced 43 polymorphic SSRs from 4,394 EST sequences of *Z. aethiopica*, representing the first dataset of polymorphic EST-SSR markers for the genus *Zantedeschia* (*Wei et al., 2012*). This conventional method of EST-SSR development has a high development cost and a low throughput, which may restrict the use of SSR markers in further genetic breeding (*Morgante, Hanafey & Powell, 2002*; *Varshney, Graner & Sorrells, 2005*).

Recent advances in RNA sequencing (RNA-seq) technology and *de novo* assembly provide an excellent strategy for the efficient and cost-effective development of high-throughput EST-SSRs. This strategy can enrich the large amounts of expressed sequence data for non-model organisms for which the reference genome or transcriptome is not available (*Martin & Wang, 2011*). Several non-model organisms in the Araceae family, including *Z. aethiopica* (*Cândido et al., 2014*) *Arum concinnatum* (*Onda et al., 2015*), *Anthurium andraeanum* (*Tian et al., 2013*; *Yang et al., 2015*; *Li et al., 2015*), *Pinellia ternate* (*Wang et al., 2014*), *Amorphophallus konjac* and *A. bulbifer* (*Zheng et al., 2013*; *Diao et al., 2014*), have been recently studied by transcriptome sequencing, which has provided a better understanding of these crops. EST-SSR markers have been successfully characterized and developed in *P. ternate* (*Wang et al., 2014*), *A. konjac* and *A. bulbifer* (*Zheng et al., 2013*). A total of 14,468 and 19,596 EST-SSRs were identified in 12,000 and 16,027 non-redundant transcriptome unigenes, respectively, of *P. ternate* and two *Amorphophallus* species (*Wang et al., 2014*; *Zheng et al., 2013*). Furthermore, 320 primers were synthesized and used to validate the amplification and assessment of polymorphisms in 25 individual *Amorphophallus* spp. plants (*Zheng et al., 2013*), of which 275 primer pairs yielded PCR amplification products and 205 were polymorphic. This strongly demonstrated that *de novo* assembly based on RNA-seq can offer a simple, direct and reliable approach for the identification and development of massive unigene-based microsatellite markers with diverse motifs. In addition to the development of molecular markers, transcriptome sequencing has also been used for the discovery, profiling and quantification of RNA transcripts and novel genes. For example, three lipid transfer proteins (LTP) that are potentially involved in defense against pathogens or predators were identified by the *in silico* screening of the 83,578 transcriptome contigs of *Z. aethiopica* (*Cândido et al., 2014*). This represents the first transcriptome dataset for the genus *Zantedeschia*.

Here, we presented the generation of a large expressed sequence dataset based on Illumina HiSeq™ 2000 sequencing data from mixed tissues of colored calla lily, *Z. rehmannii* cv. 'Rehmannii'. The objectives were as follows: (1) to characterize and annotate the transcriptome information of colored calla lily; (2) to identify all of the candidate genes encoding enzymes or putative transcription factors that are involved in its flower spathe development; (3) to analyze the frequency and distribution of SSRs and SNPs in its transcribed regions; (4) to develop a large set of EST-SSR markers; and (5) to evaluate the

transferability and genetic diversity of 21 cultivars or hybrids of colored calla lily using these EST-SSR markers.

## MATERIALS AND METHODS

### Plant materials

All of the samples used in the present study were maintained at the Bulb and Perennial Flowers Genebank (BPFG), Beijing Academy of Agriculture and Forestry Sciences, Beijing, China. The samples were grown in the experimental greenhouse in Yanqing Farm (latitude 40.46N, longitude 115.91E); this cultivation did not involve endangered or protected species. *Z. rehmannii* cv. Rehmannii, a potted cultivar with a pink spathe, was selected for transcriptome sequencing. An additional twenty accessions (Table S1), including nine individuals from New Zealand (Elmaro, Pink Diamond, Butter Gold, Black Magic, Majestic Red, Sunny Baby, Greta, Goldilocks and Gold affair), five from the Netherlands (Odessa, Captain Reno, Allure, Captain Romance, and Captain Murano), three from the United States (Parfait, Super Gem and Rose Gem) and three from China (Jingcai Yangguang, ZH and Xiangyuan Red), were used to estimate microsatellite variations of the EST-SSR markers developed in our study. All of the accessions are released commercial varieties of New Zealand, the Netherlands, the United States or China; thus, permission for collection is not required.

### RNA extraction, cDNA library construction and Illumina sequencing

To achieve a comprehensive collection of expressed transcripts of colored calla lily, ten 'Rehmannii' tissue types, including root, tuber, stem, leaf, bud, spathe, pistillate inflorescence and staminate inflorescence, immature seed and mature seed, were separately harvested at the anthesis and wilting stages. All of the tissue samples were immediately frozen in liquid nitrogen and stored at −80 °C until RNA extraction. Total RNA Trizol Reagent (Invitrogen, USA) was used to extract RNA from all of the samples, following the manufacturer's instructions. The RNA purity and concentration were determined using a NanoDrop spectrophotometer (Thermo, USA). Equal quantities of total RNA from each sample were then pooled together and used for transcriptome sequencing.

cDNA library construction and Illumina-based sequencing were performed by the Shanghai Majorbio Bio-pharm Biotechnology Co. Ltd. (Shanghai, China) according to the manufacturer's instructions. The paired-end cDNA sequencing libraries with approximate average insert lengths of 200 base pairs were prepared from the total RNA, as per the protocol of the Genomic Sample Prep kit (Illumina, San Diego, CA). This process started with mRNA fragmentation, followed by reverse transcription, first- and second-strand synthesis, paired-end adapter ligation and PCR amplification. Library quantification and quality assessment were performed on an Agilent 2100 Bioanalyzer and an ABI StepOnePlus Real-Time PCR System. Finally, the cDNA library was sequenced as 101-mer×2 on an Illumina HiSeq 2000 using paired-end sequencing chemistry.

### *De novo* transcriptome assembly and functional annotation

The raw sequencing image data were transformed into raw reads and stored in FASTQ format. These raw data were then filtered and deposited in the National Center

for Biotechnology (NCBI) Sequence Read Archive (SRA) under accession number SRR3310941. All of the reads with adaptor contamination, empty reads, non-coding RNA (such as rRNA, tRNA and miRNA), ambiguous nucleotides comprising more than 5% or low quality value (QV), i.e., an average QV of less than 20 (QV < 20), were discarded or filtered. Then, the clean and high-quality transcriptome sequence data were *de novo* assembled using the short reads assembling program Trinity (*Haas et al., 2013*) with default settings. An assembled transcripts database (File S1) was finally achieved and arranged according to the gene family clustering analysis. These final assembled transcripts were used for further bioinformatics analysis.

The assembled sequences were functionally annotated by BLASTx against a series of databases, including the Nr, Swiss-Prot, COG, and KEGG databases, with a common significance threshold cut-off of *E*-value 1e-10$^{-5}$. Based on Nr annotation, Blast2GO (https://www.blast2go.com/) and WEGO (http://wego.genomics.org.cn/cgi-bin/wegol) software were used to retrieve and classify GO annotation categories defined by molecular function, cellular component and biological process.

## Marker locus detection and SSR primer pair design

MIcroSAtellite identification tool (MISA, http://pgrc.ipk-gatersleben.de/misa/misa.html) was used to identify microsatellites in the assembled transcripts. The minimum number of repeats used to select the SSRs was ten for mononucleotide repeats, six for dinucleotide repeats, and five for tri-, tetra-, penta-, and hexanucleotide repeats. Primer Premier 5.0 (PREMIER Biosoft International, Palo Alto, CA) was then used to manually design 200 pairs of PCR primers to randomly select sequences with SSR loci. The design criteria of the primers were as follows: primer length 18–24 bp; GC content 40–65%; melting temperature 50–65 °C; and expected product size 100–300 bp with no secondary structures. All 200 primer pairs were synthesized by Sangon Biological Engineering Technology (Shanghai, China).

Potential SNPs were detected using the programs BWA (http://biobwa.sourceforge.net/) and VarScan (http://varscan.sourceforge.net/). The assembled unigenes were used as references to BLAST the raw sequencing reads.

## DNA extraction and EST-SSR marker amplification

The young leaves were collected from 21 accessions (Table S1) grown at BPFG. The total genomic DNA was extracted using the DNeasy Plant Mini Kit (Zexing Biotech, China) following the manufacturer's protocol. The quantity and quality of DNA were evaluated using a Nanodrop ND 1,000 spectrophotometer (Thermo Scientific, USA). The DNA was adjusted to a concentration of 20 ng/µl and stored at −20 °C until use.

The SSR amplification reactions were conducted as described previously by *Wei et al. (2012)* with little modification. The PCR reaction mixtures (10 µl) contained 1× PCR buffer, 20–30 ng of template DNA, 0.8 mM MgCl2, 15 µM dNTPs, 0.25 µM each primer, and 0.2 U of Taq DNA polymerase (Zexing Biotech, China). PCR amplifications were performed in the GeneAmp PCR System 9700 (Applied Biosystems). The thermal profile included an initial denaturation at 95 °C for 5 min, followed by 20 cycles of 95 °C for 30 s,

50–60 °C for 45 s, and 72 °C for 60 s and a final extension at 72 °C for 10 min. The PCR products were separated on 8.0% polyacrylamide non-denaturing gels and visualized by silver staining (*Wei et al., 2012*). The product sizes were determined by comparison with a 34- to 501-bp pUC19/MspI DNA marker (Zexing Biotech, China).

### Data analysis

The number of alleles (*Na*), observed heterozygosity (*Ho*), expected heterozygosity (*He*), and polymorphic information content (*PIC*) were calculated using GenAlEx 6.4 (*Peakall & Smous, 2006*) and Power Marker Version 3.25 (*Liu & Muse, 2005*). A cluster analysis was conducted and displayed using the neighbor-joining (NJ) algorithm as implemented in Power Marker Version 3.25 (*Liu & Muse, 2005*) and MEGA version 5.0 (*Tamura et al., 2011*).

## RESULTS

### Illumina sequencing and *de novo* assembly

To provide a comprehensive transcriptome database for calla lily, a mixed cDNA library of cultivar 'Rehmannii' from ten sampled tissues, i.e., root, tuber, stem, leaf, bud, spathe, pistillate inflorescence and staminate inflorescence, immature seed and mature seed, was constructed and sequenced using Illumina paired-end technology. A summary of the sequencing output statistics is shown in Table 1. The sequencing yielded approximately 59.9 million raw reads with a total of more than 6.1 billion nucleotides. After removing ambiguous, low-quality reads and reads with adaptors, 46.3 clean reads were obtained for further assembly. The *de novo* assembly yielded 62,382 transcripts and 39,298 unigenes. The sequences of all the assembled unigenes were provided in FASTA format in File S1. These sequences contained 40.8 Mb of sequence with an average size of 1,038 bp and an N50 of 1,476 bp. The lengths of the unigenes ranged from 351 to 15,521 bp. Of these unigenes, 67.3% (26,455) were shorter than 1,000 bp, 20.3% (7,963) ranged from 1,000 to 2,000 bp, and the remaining 12.4% (4,880) were longer than 2,000 bp (Table 1). These unigenes formed a potential pool for identification of genes and functional molecular markers in colored calla lily.

### Functional annotation of the unigenes

Functional annotations were performed by a homology-based approach for cultivars 'Rehmannii' assembled transcripts. A sequence similarity search was first conducted against the Nr and Swiss-Prot databases using the BLASTx algorithm with an *E*-value threshold of $10^{-5}$ (Table 1 and Table S2). As shown in Table 1, the results indicated that 21,029 (53.5%) of the 39,298 unigenes showed significant BLASTx matches in the Nr database, while 16,908 (43.0%) were similar to proteins in the Swiss-Prot database. The proportion of sequences showing hits in both the Nr and Swiss-Prot databases was higher among the longer assembled transcripts. More than 55.0% of the unigenes longer than 1,000 bp showed homologous matches while fewer than 45.0% of the unigenes shorter than 1,000 bp showed matches (File S1). The *E*-value, sequence similarity and species distributions of the top hits in the Nr database were also analyzed. The *E*-value distribution

**Table 1   Summary of transcriptome statistics and functional annotation for colored calla lily 'Rehmannii'.**

|  | Number | Percentage |
|---|---|---|
| Raw reads | 59,882,890 | |
| Total sizes (nt) | 6,048,171,890 | |
| Clean reads | 46,343,613 | |
| Transcripts | 62,382 | |
| Unigenes | 39,298 | |
| Unigenes (300–500 nt) | 13,367 | 34.02% |
| Unigenes (500–1,000 nt) | 13,088 | 33.30% |
| Unigenes (1,000–1,500 nt) | 4,975 | 12.65% |
| Unigenes (1,500–2,000 nt) | 2,988 | 7.60% |
| Unigenes (>2,000 nt) | 4,880 | 12.41% |
| Mean length (nt) | 1,038 | |
| N50 (nt) | 1,476 | |
| GC% | 45.74% | |
| Annotated in Nr | 21,029 | 53.51% |
| Annotated in Swiss-Prot | 16,908 | 43.03% |
| Annotated in COG | 6,731 | 17.13% |
| Annotated in GO | 15,552 | 39.57% |
| Annotated in KEGG | 4,532 | 11.53% |
| Annotated in at least one database | 21,077 | 53.63% |
| Total unigenes | 39,298 | 100% |

of the top hits showed that 62.7% of the annotated sequences had high homology with $E$-value < 10–50, whereas 37.3% showed a moderate homology with $E$-values from $10^{-5}$ to $10^{-50}$ (Fig. 1A). For the sequence similarity distribution analysis, 2,365 (11.3%), 6,902 (32.8%), 8,500 (40.4%), 3,202 (15.2%) and 45 (0.21%) sequences were 23–40%, 40–60%, 60–80%, 80–100% and 100% similar in the Nr database, respectively (Fig. 1B). In addition, the species distribution showed that *Vitis vinifera* (Vitaceae) was ranked first, with 6,720 (32.0%) top BLASTx hits, followed by *Theobroma cacao* (Sterculiaceae), *Prunus persica* (Rosaceae), *Ricinus communis* (Euphorbiaceae) and *Populus trichocarpa* (Salicaceae), with 2,117 (10.1%), 1,176 (5.6%), 1,034 (4.9%), and 1,014 (4.8%) hits, respectively (Fig. 1C).

## Functional classification of the unigenes

All of the assembled unigenes were subjected to a search against the COG, GO and KEGG databases for further functional prediction and classification (Table S2). The COG database can provide phylogenetic classification of proteins encoded by several complete genomes of bacteria, archaea and eukaryotes (*Roman et al., 2000*). The COG function classification of the 'Rehmannii' sequences is shown in Fig. 2A. In total, 6,731 of the 39,298 unigenes showing Nr hits were functionally annotated and classified into 24 COG categories, including cellular structure, biochemistry metabolism, molecular processing, and signal transduction, among others. Given that 9,379 COG-annotated putative proteins were obtained, some of these unigenes were assigned to multiple COG classifications. The cluster for general function prediction only (1,791, 26.6%) represented the largest group,

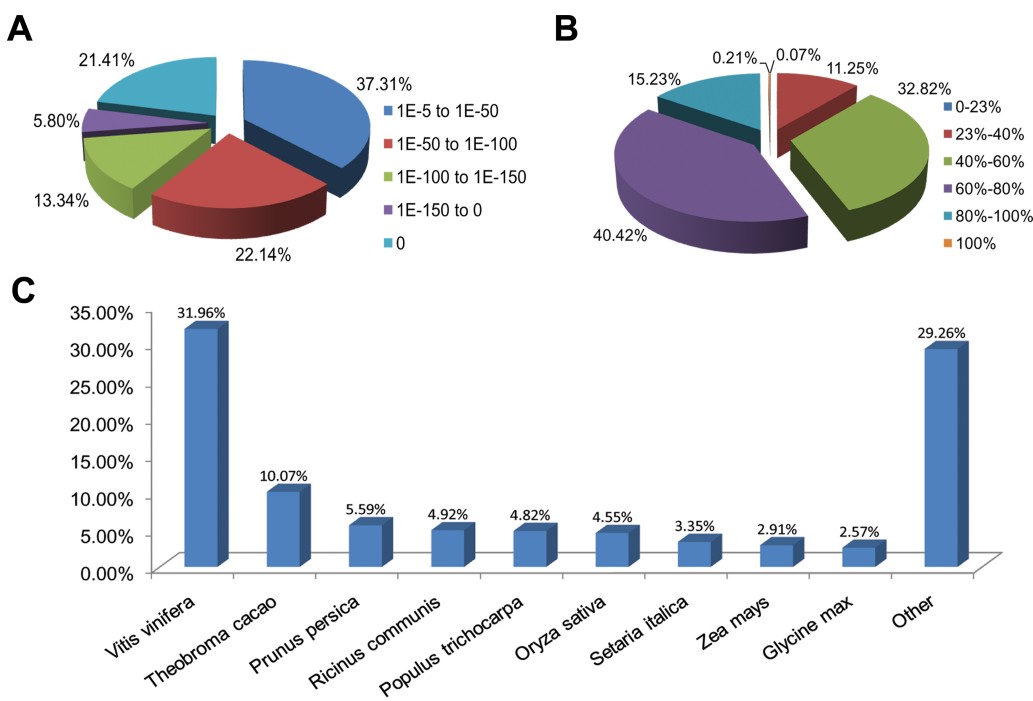

**Figure 1  Characteristics of homology search for colored calla lily 'Rehmannii' unigenes against non-redundant protein database (Nr) with an *E*-value = 1E-10⁻⁵.** (A) The *E*-value distribution of BLASTx hit for each assembled unigene; (B) the similarity distribution of BLASTx hits for each assembled unigenes; (C) species-based distribution of the top BLASTx hits for each assembled unigenes.

followed by replication, recombination and repair (904, 13.4%) and transcription (836, 12.4%). Additionally, only a few unigenes were assigned to cell motility (9, 0.13%) and nuclear structure (1, 0.01%).

The GO database, an internationally standardized gene functional classification system, offers dynamic and updated gene ontology that defines gene products in terms of their associated cellular component, molecular function, and biological process in any organism (*Ashburner et al., 2000*). Based on Nr annotation, 15,552 (39.6%), unigenes in the present study were assigned to GO classes with 15,964 functional terms. A summary of the 'Rehmannii' unigenes classified to each GO Slim term is shown in Fig. 2B. The annotated gene sequences that belong to the cellular component, molecular function, and biological process categories were divided into 47 functional groups. Under the cellular component category, cell (4,254, 27.4%) and cell part (4,253, 27.4%) were the most highly represented group, followed by organelle (3,331, 21.4%). For the molecular function category, the top two groups were binding (3,512, 22.6%) and catalytic activity (3,313, 21.3%). However, the majority of the groups, including antioxidant activity, receptor activity, protein binding transcription factor activity, nutrient reservoir activity, etc., contained only a few unigenes (655, 4.2%). For biological processes, the majority of the unigenes were involved in cellular processes (2,344, 15.1%) and metabolic processes (1,814, 11.7%), indicating that important metabolic and cellular activities occur in 'Rehmannii'. Genes involved in other important biological processes, such as single-organism processes (1,337, 8.6%), stimulus response

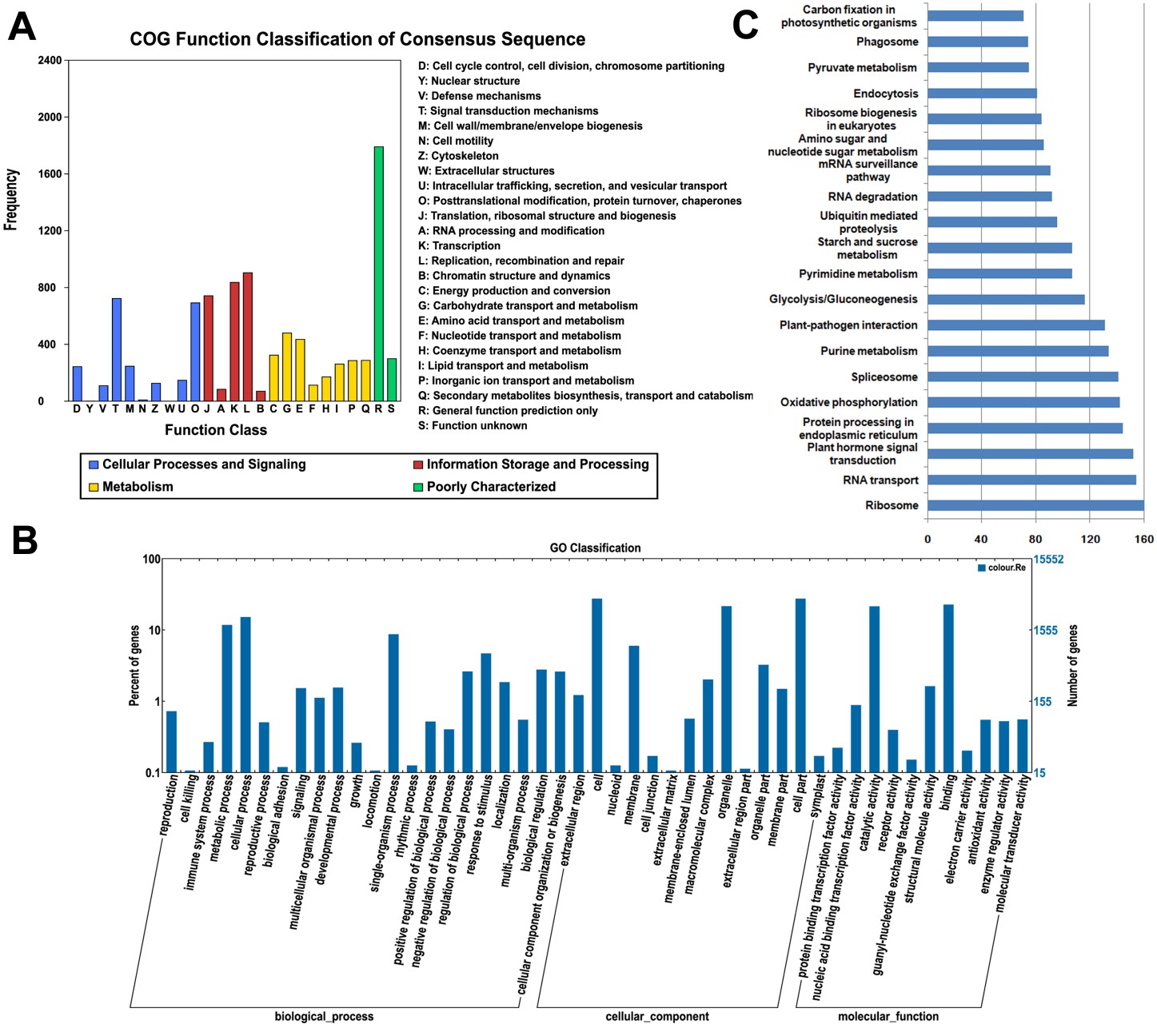

**Figure 2** **The classification of colored calla lily 'Rehmannii' unigenes.** (A) Distribution of Cluster of Orthologous Group (COG) classification. A total of 6,731 assembled unigenes were annotated and assigned to 24 functional categories. The *x*-axis indicates the subgroups in the COG classification while the *y*-axis indicates the number of genes in each main category. (B) Gene ontology (GO) classification of assembled unigenes at level 2. A total of 15,552 unigenes were grouped into three main GO categories: 'Biological Processes', 'Cellular Component', and 'Molecular Function'. The *x*-axis indicates the subgroups in GO annotation while the *y*-axis indicates the percentage of specific categories of genes in each main category. (C) The top 20 KEGG metabolic pathways of assembled unigenes. The *x*-axis indicates the number of genes in each metabolic pathway while the *y*-axis indicates the subgroups in the top 20 KEGG metabolic pathways.

(712, 4.6%) and biological regulation (416, 2.7%), also accounted for a large fraction of the annotated transcripts. Furthermore, we found that a portion of the unigenes were assigned to developmental processes (226, 1.5%), reproductive processes (63, 0.4%) and growth (25, 0.2%) and may be involved in flower-related biological processes, such as flower-type development and formation, in colored calla lily.

The KEGG database (*Kanehisa et al., 2004*) determines the biological pathways in which protein and small-molecule interactions occur. A total of 4,532 assembled sequences (11.5%) were consequently annotated and assigned to 117 predicted metabolic pathways. The number of sequences in each pathway ranged from 1 to 160. Fig. 2C shows the top 20 KEGG metabolic pathways represented by unique transcripts of calla lily. Ribosome represented the largest number of transcript sequences in our dataset (160), followed by RNA transport (154) and plant hormone signal transduction (152). Among the annotated sequences, more than one-third (1,661) were classified into 46 metabolic groups, such as purine metabolism (134), pyrimidine metabolism (107), starch and sucrose metabolism (75), and so on. This indicates that diverse metabolic processes are active and a variety of metabolites are synthesized in the tissues and organs of colored calla lily.

Overall, 21,077 unique sequence-based annotations using the selected Nr, Swiss-Prot, COG, GO and KEGG databases were assigned to assembled transcripts of the cultivar 'Rehmannii'. The functional analysis revealed that it is feasible to obtain transcriptome sequences through high-throughput technology, even for non-model plants with large genomes. Furthermore, all of these functional annotation assignments provide valuable information for colored calla lily to investigate specific biochemical and developmental processes and potential gene structures, functions and pathways.

## Unigenes related to flower development in colored calla lily

Enzymes functioning in flower development have been well documented in many plants. Based on Nr annotation results of colored calla lily unigenes, we identified a total of 171 candidate transcripts encoding enzymes related to pigment biosynthesis (61), floral organ development (26), flowering regulation (63) and flower senescence (31). The EST sequences of all these unigenes are listed in Table S3. Flower development is a complex process controlled by an integrated network of multi-genetic pathways in higher plants. The putative gene homologs identified in the present study were involved in eight pathways, including the anthocyanin biosynthesis pathway (52), carotenoid biosynthesis pathway (9), specification of floral organ identity (26) , photoperiod pathway (22), vernalization pathway (11), gibberellic acid pathway (11), autonomous or other pathways (9) and ethylene biosynthesis pathway (31). The identification and analysis of these key genes will provide a foundation for understanding the potential molecular genetic mechanisms controlling different aspects of floral development of colored calla lily in the future.

## SSR and SNP loci discovery

Transcriptome sequencing is important for identification and development of molecular markers, such as SSRs and SNPs. For the development of new markers for colored calla lily, all of the 39,298 unigenes generated in this study were used to mine potential microsatellites

**Table 2  Features of the SSR repeat types identified in colored calla lily 'Rehmannii' unigenes.**

| Feature | Colored calla lily |
|---|---|
| Total number of sequences examined | 39,298 |
| Total size of examined sequences (Mb) | 40.78 |
| Total number of identified SSRs | 9,933 |
| Number of SSR-containing sequences | 7,997 |
| Number of sequences containing more than one SSR locus | 1,556 |
| Number of SSRs present in compound formation | 580 |

using MISA software. A total of 9,933 potential EST-SSRs were identified in 7,997 unigenes, of which 1,556 sequences contained more than one SSR, and 580 SSRs were present in compound form (Table 2 and Table S4). Considering that approximately 40,780 kb was analyzed, we detected a frequency of at least one SSR per 4.1 kb in the expressed fraction of the 'Rehmannii' genome.

The type and frequency of EST-SSRs with different numbers of tandem repeats are summarized in Table 3. Because mononucleotide repeats may not be accurate due to sequencing errors and assembly mistakes, they were excluded from further analyses. Results showed that the identified SSR type was not evenly distributed throughout all the SSR-containing sequences of 'Rehmannii'. The dinucleotide repeat motifs were the most abundant (3,482 or 59.8%), followed by trinucleotide repeat motifs (2, 261 or 38.8%), whereas hexa- (62 or 1.06%), penta- (13 or 0.22%) and tetranucleotide repeat motifs (7 or 0.12%) were rare. The number of SSR repeats ranged from 5 to 24, with 6 repeats (1,477, 25.4%) being the most common, followed by 5 tandem repeats (1,465, 25.2%) and 7 tandem repeats (981, 16.8%). Motifs containing more than 10 repeats were rare (Table S2), with a frequency of only 2.6% (151). Within these SSRs, 47 motif sequence types were identified, of which di-, tri-, tetra-, penta-, and hexanucleotide repeats had 4, 10, 13, 13 and 7 types, respectively. AG/CT alone accounted for 79.5% (2,768) of the total dinucleotide repeats, followed by AT/TA (360,10.4%) and AC/GT (328, 9.4%). Among the trinucleotide repeats, GAA/CTT and AAG/CTT were the most abundant (480, 21.2%; 468, 20.7%). Other repeats, AGC/CTG, CCG/CGG, ATC/ATG and ACC/GGT, constituted 47.4% of the trinucleotide repeats.

In addition to EST-SSRs, a total of 7,162 potential high-quality SNPs were identified by mapping against 39,298 reference unigenes. The overall frequency of all types of SNPs was one SNP per 5.69 kb. The predicted SNPs included 4,450 transitions and 2,712 transversions (Table 4). The most abundant SNPs detected were C/T (2,262, 31.6%), followed by A/G (2,188, 30.6%) and C/G (790, 11.0%). The numbers of the remaining three SNP types (A/T, A/C, and T/G) were similar, each accounting for less than 10%. Potential SNPs are shown in Table S5.

## SSR primer design, polymorphism detection and phylogenetic analysis

A total of 200 EST-SSR loci (repeat motif >1) with appropriate flanking sequences were randomly selected for the design and synthesis of PCR primer pairs. Detailed information

**Table 3  Summary of EST-SSRs identified from the unigenes of colored calla lily 'Rehmannii'.**

| Repeat motif | Number of repeats | | | | | | | |
|---|---|---|---|---|---|---|---|---|
| | 5 | 6 | 7 | 8 | 9 | 10 | >10 | Total |
| Di- (3,482, 59.78%) | | | | | | | | |
| AG/CT | 0 | 652 | 568 | 525 | 536 | 381 | 106 | 2,768 |
| AT/AT | 0 | 108 | 68 | 61 | 46 | 49 | 28 | 360 |
| AC/GT | 0 | 132 | 70 | 54 | 25 | 32 | 15 | 328 |
| CG/CG | 0 | 14 | 7 | 0 | 3 | 2 | 0 | 26 |
| Tri- (2,261, 38.82%) | | | | | | | | |
| AGG/CCT | 316 | 116 | 43 | 4 | 1 | 0 | 0 | 480 |
| AAG/CTT | 258 | 136 | 67 | 6 | 0 | 0 | 1 | 468 |
| AGC/CTG | 252 | 99 | 39 | 2 | 0 | 0 | 0 | 392 |
| CCG/CGG | 245 | 85 | 39 | 3 | 0 | 0 | 0 | 372 |
| ATC/ATG | 96 | 37 | 21 | 3 | 0 | 0 | 0 | 157 |
| ACC/GGT | 84 | 42 | 20 | 3 | 1 | 0 | 0 | 150 |
| Other | 146 | 49 | 35 | 11 | 1 | 0 | 0 | 242 |
| Tetra- (62, 1.06%) | | | | | | | | |
| AAAG/CTTT | 14 | 2 | 0 | 0 | 0 | 0 | 0 | 16 |
| AGAT/ATCT | 12 | 1 | 0 | 0 | 0 | 0 | 0 | 13 |
| ACAT/ATGT | 4 | 1 | 1 | 0 | 0 | 0 | 0 | 6 |
| AAAT/ATTT | 5 | 0 | 0 | 0 | 0 | 0 | 0 | 5 |
| Others | 17 | 3 | 1 | 1 | 0 | 0 | 0 | 22 |
| Penta- (13,0.22%) | 13 | 0 | 0 | 0 | 0 | 0 | 0 | 13 |
| Hexa- (7, 0.12%) | 3 | 0 | 2 | 1 | 0 | 0 | 1 | 7 |
| Total | 1,465 | 1,477 | 981 | 674 | 613 | 464 | 151 | 5,825 |
| Percentage | 25.15% | 25.36% | 16.84% | 11.57% | 10.52% | 7.97% | 2.59% | 100% |

**Table 4  Summary of SNPs identified from unigenes of colored calla lily 'Rehmannii'.**

| Transitions | Number | Transversions | Number |
|---|---|---|---|
| C/T | 2,262 | A/T | 650 |
| A/G | 2,188 | A/C | 647 |
| | | T/G | 625 |
| | | C/G | 790 |
| Total | 4,450 | Total | 2,712 |

of the EST-SSR markers is shown in Table S6. A germplasm panel of five colored calla lily accessions (Rehmannii, Super Gem, Rose Gem, Xiangyuan Red and Allure) was initially used to validate the usefulness of EST-SSR markers in monitoring polymorphisms. A total of 137 (68.5%) of the primer pairs were successfully amplified by PCR, while the remaining failed to generate any clear DNA products. Of the working primer pairs, 77 (56.2%) produced clear PCR amplicons of the expected sizes, whereas 60 (43.8%) amplified non-specific products, of which 23 markers generated PCR products larger or smaller than expected and 37 generated more than one band (Table S4).

**Table 5  Characteristics of the 58 polymorphic EST-SSR markers in 21 colored calla lily accessions.**

| Locus | Na | Ne | Ho | He | PIC | Locus | Na | Ne | Ho | He | PIC |
|-------|-----|------|------|------|------|-------|-----|------|------|------|------|
| CallaRe015 | 3 | 2.057 | 0.684 | 0.514 | 0.425 | CallaRe110 | 2 | 1.324 | 0.286 | 0.245 | 0.215 |
| CallaRe016 | 6 | 3.320 | 0.600 | 0.699 | 0.654 | CallaRe117 | 3 | 2.410 | 0.000 | 0.585 | 0.513 |
| CallaRe028 | 4 | 3.756 | 1.000 | 0.734 | 0.685 | CallaRe118 | 4 | 2.028 | 0.667 | 0.507 | 0.462 |
| CallaRe030 | 3 | 2.095 | 1.000 | 0.523 | 0.409 | CallaRe120 | 4 | 2.932 | 0.333 | 0.659 | 0.593 |
| CallaRe031 | 2 | 1.446 | 0.286 | 0.308 | 0.261 | CallaRe128 | 4 | 3.556 | 0.550 | 0.719 | 0.670 |
| CallaRe032 | 2 | 1.930 | 0.810 | 0.482 | 0.366 | CallaRe129 | 2 | 1.984 | 0.545 | 0.496 | 0.373 |
| CallaRe036 | 2 | 1.296 | 0.263 | 0.229 | 0.202 | CallaRe131 | 4 | 2.930 | 0.286 | 0.659 | 0.601 |
| CallaRe040 | 4 | 2.139 | 0.278 | 0.532 | 0.483 | CallaRe135 | 2 | 1.220 | 0.200 | 0.180 | 0.164 |
| CallaRe041 | 3 | 1.156 | 0.048 | 0.135 | 0.130 | CallaRe144 | 4 | 2.766 | 0.588 | 0.638 | 0.589 |
| CallaRe042 | 2 | 1.265 | 0.238 | 0.210 | 0.188 | CallaRe146 | 3 | 2.085 | 0.952 | 0.520 | 0.408 |
| CallaRe049 | 2 | 1.960 | 0.857 | 0.490 | 0.370 | CallaRe147 | 3 | 1.841 | 0.353 | 0.457 | 0.411 |
| CallaRe050 | 6 | 3.630 | 0.571 | 0.724 | 0.683 | CallaRe151 | 3 | 2.455 | 0.667 | 0.593 | 0.505 |
| CallaRe055 | 3 | 2.057 | 0.526 | 0.514 | 0.425 | CallaRe155 | 2 | 1.724 | 0.000 | 0.420 | 0.332 |
| CallaRe056 | 2 | 1.946 | 0.833 | 0.486 | 0.368 | CallaRe156 | 4 | 2.309 | 0.667 | 0.567 | 0.486 |
| CallaRe061 | 4 | 3.469 | 0.286 | 0.712 | 0.661 | CallaRe160 | 2 | 1.893 | 0.000 | 0.472 | 0.360 |
| CallaRe066 | 2 | 1.265 | 0.238 | 0.210 | 0.188 | CallaRe165 | 2 | 2.000 | 1.000 | 0.412 | 0.375 |
| CallaRe075 | 2 | 1.358 | 0.313 | 0.264 | 0.229 | CallaRe166 | 4 | 2.520 | 0.333 | 0.603 | 0.541 |
| CallaRe078 | 4 | 2.303 | 0.952 | 0.566 | 0.471 | CallaRe170 | 3 | 2.597 | 0.737 | 0.615 | 0.536 |
| CallaRe080 | 4 | 2.285 | 0.381 | 0.562 | 0.519 | CallaRe175 | 6 | 4.762 | 0.800 | 0.790 | 0.757 |
| CallaRe081 | 3 | 1.407 | 0.333 | 0.289 | 0.266 | CallaRe178 | 2 | 1.835 | 0.700 | 0.455 | 0.351 |
| CallaRe082 | 3 | 2.182 | 0.000 | 0.542 | 0.460 | CallaRe179 | 2 | 1.992 | 0.188 | 0.498 | 0.374 |
| CallaRe089 | 2 | 1.995 | 0.857 | 0.499 | 0.374 | CallaRe180 | 3 | 1.340 | 0.190 | 0.254 | 0.237 |
| CallaRe090 | 3 | 2.829 | 0.500 | 0.646 | 0.571 | CallaRe185 | 2 | 1.637 | 0.412 | 0.389 | 0.314 |
| CallaRe095 | 3 | 2.524 | 1.000 | 0.604 | 0.525 | CallaRe187 | 2 | 1.960 | 0.000 | 0.490 | 0.370 |
| CallaRe097 | 3 | 1.956 | 0.619 | 0.489 | 0.407 | CallaRe189 | 3 | 2.111 | 0.550 | 0.526 | 0.431 |
| CallaRe100 | 3 | 2.246 | 0.500 | 0.555 | 0.456 | CallaRe190 | 2 | 1.498 | 0.316 | 0.332 | 0.277 |
| CallaRe101 | 3 | 1.893 | 0.095 | 0.472 | 0.397 | CallaRe191 | 2 | 1.205 | 0.188 | 0.170 | 0.155 |
| CallaRe106 | 4 | 2.431 | 0.684 | 0.589 | 0.506 | CallaRe194 | 2 | 1.600 | 0.500 | 0.375 | 0.305 |
| CallaRe109 | 3 | 2.256 | 0.619 | 0.557 | 0.462 | CallaRe198 | 3 | 2.492 | 0.650 | 0.599 | 0.514 |

Specific EST-SSR markers were used to assess the genetic diversity and relationships among the 21 accessions of colored calla lily from New Zealand, the Netherlands, the United States and China. Of these tested markers, 58 (or 75.3%) were polymorphic, while the others were monomorphic. The polymorphic EST-SSRs consisted of 22 di-, 18 tri-, 6 tetra-, 4 penta-motif- and 8 compound-motif-based markers. The raw data and characteristics of the 58 polymorphic EST-SSR markers in 21 accessions are listed in Table S7 and Table 5. The number of alleles per marker ($Na$) ranged from 2 to 6, with 174 alleles in total. The average effective number of alleles per locus ($Ne$) was 2.163, with a maximum of 4.762 and a minimum of 1.156. The observed heterozygosity ($Ho$) varied from 0 to 1.000, whereas the expected heterozygosity ($He$) varied from 0.135 to 0.790. The mean $Ho$ and $He$ values were 0.483 and 0.491, respectively. The polymorphic information content ($PIC$) values ranged from 0.130 to 0.757 with a mean value of 0.420. Nr annotation results (Table S6) showed

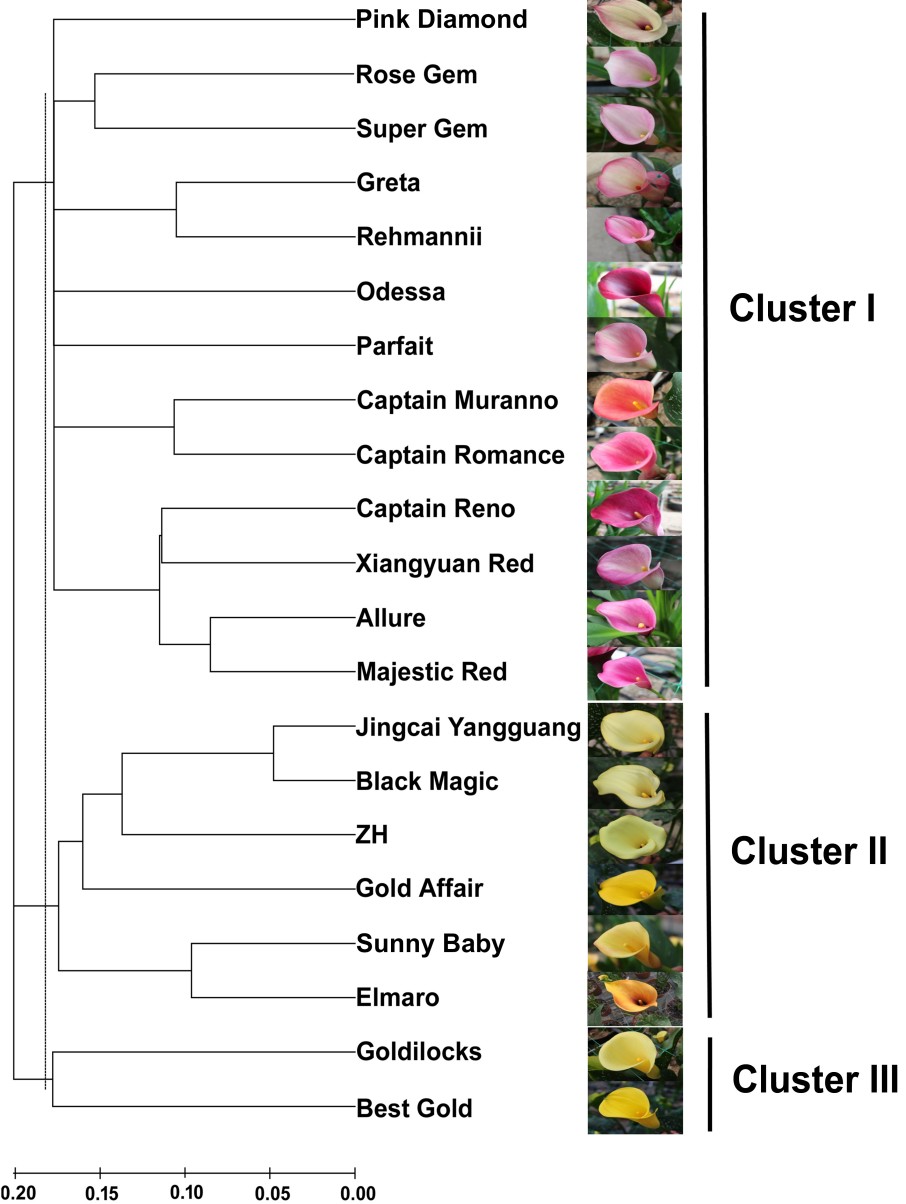

**Figure 3** An NJ dendrogram of 21 colored calla lily accessions based on 58 polymorphic EST-SSR markers.

that most of the polymorphic SSR-containing sequences shared significant homology to protein-encoding genes from *V. vinifera* (Vitaceae), *T. cacao* (Sterculiaceae), *Z. mays* and *O. sativa* Indica (Gramineae), among others. These positive-hit homologous genes, including nucleic acid binding protein, glycine-rich protein, abscisic acid, zinc-finger transcription factor, had hypothetical or putative functions in defense and stress, transporters and metabolic processes.

Based on the shared allele distance, we conducted a cluster analysis to assess the genetic relationships between 21 accessions of colored calla lily (Fig. 3). The dendrogram clearly

grouped these accessions into three major clusters (I, II and III) at an approximate genetic distance of 0.18. Cluster I consisted of 13 accessions collected from the Netherlands, the United States, New Zealand and China. Twelve accessions in clusters I (Pink Diamond, Rose Gem, Super Gem, Greta, Parfait, Rehmannii, Odessa, Majestic Red, Xiangyuan Red, Captain Reno, Allure, and Captain Romance) had red, purple and pink spathes, whereas Captain Murano had an orange spathe. Clusters II and III consisted of eight yellow-flower accessions. The accessions in the former cluster (Jingcai Yangguang, Black Magic, ZH, Gold affair, Sunny Baby and Elmaro) originated from New Zealand and China, while those in the latter (Goldilocks and Butter Gold) were collected from New Zealand. The results may indicate a potential association between the genetic relationship of the tested accessions and the spathe color.

## DISCUSSION

### Illumina transcriptome sequencing and *de novo* assembly

*Zantedeschia* spp. are perennial bulb plants known worldwide for their ornamental uses. These plants are even commonly used in traditional medicine in some African regions (*Letty, 1973*; *Singh, 1996*). However, little genomic information is available for the plants of the genus *Zantedeschia*, even with significant advances in DNA sequencing technologies. The genomic researches of *Zantedeschia* spp. may be hindered by the large genome size of these plants. The nuclear DNA contents of colored calla lily *Z. elliottiana* and white calla lily *Z. aethiopica* (*Ghimire et al., 2012*) were estimated to be $1.17 \pm 0.50$ and $3.72 \pm 0.10$ pg DNA/2C, respectively, equivalent to genome sizes of 1.15 and 3.64 Gbp, approximately 2.5 and 7.8 times larger than the rice (*O. sativa*) genome of 0.47 Gb. Moreover, *Zantedeschia* spp. may be a genetically high-heterozygosity bulb-flower species. Since these species having been bred for more than one century, a larger number of outstanding cultivars and hybrids of white and colored calla lilies in a range of sizes and colors have been cross-bred, named and released by European and American breeders. Thus, it is obviously not feasible to consider whole-genome sequencing for this perennial plant because of the high cost and time consumption.

For organisms with large, heterozygous and complex genomes containing repetitive sequences, RNA-seq-based *de novo* transcriptome analysis is an attractive alternative to examine the properties of a transcriptome as a proxy for the whole genome (*Martin & Wang, 2011*; *Onda et al., 2015*). A transcriptome study of *Z. aethiopica* spathe (*Cândido et al., 2014*) also demonstrated that transcriptome sequencing is a useful molecular biology tool to rapidly build comprehensive sequence resources of expressed genes for in-depth analysis at high resolution. To date, *de novo* transcriptome analysis has successfully been applied to a wide range of plant species (*Martin & Wang, 2011*), including crops, fruits, vegetables, forest plants, flowers, and medicinal plants, for various purposes, such as gaining fundamental insight into biological processes, generating different dynamic views of interesting gene expression, discovering novel genes or transcripts, and developing and validating molecular markers. In the present study, Illumina paired-end RNA-seq technology was used to sequence the pooled RNA from ten tissues of 'Rehmannii'. More

than 59.9 million 101-bp paired-end reads were yielded, encompassing 6.1 G nucleotides. Trinity software (*Haas et al., 2013*) was then used to *de novo* assemble these short reads to generate a total of 39,298 unigenes with an average length of 1,038 bp. The results indicated that *de novo* transcriptome assembly is a feasible strategy to provide genome information for non-model plant species. In sequence assembly, the N50 value is an important statistical measure that is used to evaluate transcriptome assemblies in which a high number corresponds to high quality. The N50 length of the 'Rehmannii' unigenes was 1,476 bp, which is comparable to a recent report of the assembled transcriptome sequences of *Z. aethiopica* (N50 = 1,600 bp) using a similar method (*Cândido et al., 2014*). However, it should be noted that *Z. aethiopica* has approximately 1.5 times (91.2 million) as many short reads as does 'Rehmannii'. These values are also larger than those reported for related species in the Araceae family, such as *A. andraeanum* (N50 = 1,172 bp), *A. konjac* and *A. bulbifer* (N50 = 381 and 534; N50 = 372 and 524, respectively) (*Zheng et al., 2013*; *Diao et al., 2014*). Previous reports have demonstrated that an accurate and effective assembly tends to have a longer mean length and a larger N50 value (*Chen et al., 2015*), which suggests the high quality of our colored calla lily transcriptome sequences.

## Functional annotation and classification of unigenes

To predict the biological functions of as many assembled transcripts as possible, various protein databases, including Nr, Swiss-Prot, COG, GO and KEGG, were employed. In total, 21,077 unigenes (Table S2) showed significant hits in the above five databases. The annotation rate of 'Rehmannii' unigenes was 53.6%, which was higher than that of *Z. aethiopica* (39.7%) using same method. The higher percentage in this study may be attributed to the higher frequency of unigenes longer than 500 bp in the assembled transcripts. In general, the longer unigenes were more likely to have BLAST matches in the protein databases (*Parchman et al., 2010*). It was estimated that the percent of unigenes >500 bp in 'Rehmannii' and *Z. aethiopica* was 65.9% and 42.1%, respectively. *Novaes et al. (2008)* reported that a high number of next-generation sequencing (NGS) short reads often cannot be matched to known genes because the significance of sequence similarity partially depends on the length of the query sequence. Therefore, the non-significantly annotated unigenes that are likely novel transcripts unique to *Z. rehmannii* 'Rehmannii' may be too short to allow for statistically meaningful matches.

A species-based distribution of the best hits from the BlastX search against the Nr database showed that 32.0% of the annotated sequences of 'Rehmannii' had similarity with dicotyledonous grape (*V. vinifera*). This finding was confirmed by other transcriptome reports on *Z. aethiopica* (*Cândido et al., 2014*), *A. andraeanum* (*Yang et al., 2015*; *Li et al., 2015*), *A. konjac* and *A. bulbifer* (*Zheng et al., 2013*). In fact, calla lily and grape are a monocotyledonous herb and a dicotyledonous woody vine, respectively. So they are distant from each other genetically and evolutionarily. One possible reason for this perceived similarity is the absence of whole-genome sequences in public databases for any species of Araceae. These assembled transcriptome sequences may provide an important data resource for future studies on taxa-specific phenomena in the family Araceae. Consistent with previous studies (*Cândido et al., 2014*; *Yang et al., 2015*; *Li et al., 2015*;

*Zheng et al., 2013*), the annotated unigenes were classified into 24 COG and 47 GO sub-terms or sub-categories, indicating that our transcriptome data represented a broad diversity of transcripts in colored calla lily. In addition, a total of 4,532 unigenes were annotated and mapped to 117 KEGG pathways. Approximately 80% of top 20 hit pathways were involved in genetic information processing and metabolism, while the others were related to pathways involved in plant hormone signal transduction, plant-pathogen interaction, phagosome, and endocytosis (Fig. 2C). The most highly represented pathway was related to genetic information processing and metabolism, reflecting the fact that calla lily devotes an enormous investment to gene transcription control and capacity, cell maintenance and defense capacity (*Cândido et al., 2014*). Unigene annotation, together with the predicted pathways, also facilitated the discovery of some key genes involved in flower development and function in colored calla lily. In total, we identified 117 homologous sequences involved in eight pathways, including the anthocyanin biosynthesis pathway, carotenoid biosynthesis pathway, photoperiod pathway, vernalization pathway, gibberellic acid pathway, autonomous or others. These captured unigenes again demonstrated that a relatively accurate and high-coverage genomic database can be produced by RNA-seq-based *de novo* transcriptome analysis for non-model plant species.

## Identification of EST-SSR marker frequency and type

Markers based on expressed sequences are useful and attractive for the detection of functional variation and gene-based analysis. However, the available EST-SSR markers are insufficient for the genus *Zantedeschia* at the present time. *Wei et al. (2012)* identified 209 EST-SSRs from 2,175 non-redundant ESTs derived from cDNA libraries of developing spathe in *Z. aethiopica*. Among these, a total of 166 primer pairs flanking the EST-SSRs could be designed. Note here that these EST-SSRs were identified based on Sanger sequencing. Very limited expressed sequence data can be generally produced via the Sanger sequencing of cDNA libraries compared to RNA-based transcriptome sequencing. In our study, 9,933 potential EST-SSRs were identified in the *de novo* transcriptome sequences obtained by Illumina sequencing (Table 2). These data confirm that transcriptome sequences are excellent resources for the development of numerous SSR markers. In the present study, it was estimated that approximately 20.3% (7,997) of the assembled unigenes possess SSR loci, and the abundance of SSRs was one SSR locus per 4.1 kb. The density of SSR-containing sequences in 'Rehmannii' was higher than that in the transcriptome reports for other species in Araceae, such as *Amorphophallus* spp. (11.8%) and *P. ternate* (16.24%) (*Wang et al., 2014*; *Zheng et al., 2013*). The frequency of SSRs was higher than that in *P. ternate* (4.3 kb) (*Zheng et al., 2013*) but lower than that in *Amorphophallus* spp. (3.6 kb) (*Wang et al., 2014*). The difference in the abundance estimation and frequency of SSRs among various species could partially be due to the SSR search criteria, the size of the unigene assembly dataset, the database-mining tools and the sequence redundancy, in addition to actual differences between species (*Wang et al., 2014*; *Zheng et al., 2013*; *Chen et al., 2015*).

The SSRs identified in this study were not uniformly distributed in the 'Rehmannii' transcriptome database (Table 3). When mononucleotide repeats were excluded, di- (59.8%) and trinucleotide repeats (38.8%) were the most abundant, whereas hexa- (1.1%),

penta- (0.2%) and tetranucleotide repeats (0.1%) were rare. This result is identical to the previous findings of di- and trinucleotide motifs as the most frequent SSR motif types in the transcriptome sequences of many other plants, including *Amorphophallus* spp. and *P. ternate* (*Wang et al., 2014*; *Zheng et al., 2013*). As shown in Table 3, the AG/CT motif was the most abundant dinucleotide repeat (27.9%), followed by AT/TA (3.6%) and AC/GT (3.3%). The predominant AG/CT motif repeats were also observed in *Amorphophallus* spp. and *P. ternate* (*Wang et al., 2014*; *Zheng et al., 2013*). The most abundant trinucleotide repeat motif in calla lily was AGG/CCT, closely followed by AAG/CTT, similar to reports in *Amorphophallus* spp. but different from those in *P. ternate* (*Wang et al., 2014*; *Zheng et al., 2013*). This difference possibly arises from the SSR search parameters and search algorithms. CCG/CGG was the most frequent trinucleotide motif in *P. ternate* but the third and fourth most common repeat type in *Amorphophallus* spp. and colored calla lily 'Rehmannii', respectively. Anyhow, these abundance results corroborated with the suggestion that the trinucleotide motif CCG/CGG is common in monocots. In addition, we also noticed that GC-rich trinucleotide motifs (ACC/GGT, ACG/CGT, AGC/CTG, AGG/CCT and CCG/CGG comprised > 65%) were more abundant than AT-rich trinucleotide motifs (AAG/CTT, AAT/ATT, ACT/AGT, and ATC/ATG comprised < 35%) in these three Araceae species. These results strongly support the fact that the high GC content and consequent codon usage bias are specific features of monocot genomes (*Morgante, Hanafey & Powell, 2002*).

## Evaluation of genetic diversity and relationships among colored calla lily accessions

Cultivar 'Rehmannii' in the present study is a hybrid of *Z. rehmannii*. It has the desired attributes for potted flower purposes, such as pink spathes, perfect trumpet-shaped inflorescence, lanceolate and semi-erect leaves, and high productivity. Similar characteristics were also observed in another four cultivars, including Super Gem, Rose Gem, Xiangyuan Red and Allure. Hence, all five varieties here were initially used to evaluate EST-SSR marker usefulness in monitoring polymorphisms. A total of 200 primer pairs were synthesized and tested, of which 137 (68.5%) successfully yielded amplicons in these five cultivars. Of the abovementioned working primer pairs, 77 (56.2%) produced PCR products of the expected fragment size used to screen for polymorphisms among 21 individual accessions. Finally, 58 polymorphic EST-SSR markers were obtained with a polymorphic proportion of 75.3%. EST-SSR markers are advantageous to SSRs in non-transcribed regions due to their higher amplification rates and cross-species transferability (*Varshney, Graner & Sorrells, 2005*). The results here indicate that these transcript-based SSRs are conserved in the germplasms of colored calla lily, suggesting that they will have a broad utilization in taxonomic and cultivar identification, as well as comparative mapping. The amplification and polymorphic rate of the EST-SSRs developed in our study (59.3%) is much higher than that obtained in *Z. aethiopica* using Sanger sequencing (*Wei et al., 2012*), suggesting that the *de novo* transcriptome sequence based on Illumina RNA-seq was accurate and of high quality. In *Z. aethiopica* (*Wei et al., 2012*), 68 (40.9%) of the EST-SSR primer pairs yielded PCR amplification products in 24 accessions, of which 43

(63.2%) exhibited polymorphisms. However, the result is lower than that reported in *Amorphophallus* spp. (*Zheng et al., 2013*). In *A. konjac* and *A. bulbifer*, 270 (84.4%) primer pairs produced amplicons in two wild accessions, and 205 (89.1%) EST-SSR markers were polymorphic between 25 wild and cultivated accessions. The low amplification and polymorphism rate in colored calla lily may be attributed to various factors (*Varshney et al., 2007*; *Varshney, Graner & Sorrells, 2005*), including the presence of introns in the corresponding cDNA, SNPs or InDels (insertion-deletions) in the primers, assembly errors in the *de novo* transcriptome sequences, and the high heterozygosity of the calla lily genome. Nonetheless, the failed amplification may be remedied if the PCR amplification conditions are re-optimized, such as by applying a lower annealing temperature and/or using gradient PCR.

Most calla lily cultivars have been bred following the intra/inter-specific hybridization of the species *Z. rehmannii*, *Z. albomaculata*, *Z. elliottiana* and *Z. pentlandii* within the section *Aestivae*, producing plants and flowers with a broad range of shapes and colors. The species *Z. rehmannii*, *Z. albomaculata*, *Z. elliottiana* and *Z. pentlandii* are closely related to each other based on cytogenetic karyotypes (*Yao, Cohen & Rowl, 1994*). However, the genetic relationship among commercial hybrids or varieties is still not well defined. Characters based on which species or even varieties have been previously separated, for example, the degree of spotting on the leaf, the presence of bristles on the petioles and peduncles and, to some extent, the leaf shape, have been unreliable when a wide range of hybrids is examined (*Letty, 1973*; *Singh, 1996*). Molecular markers, such as RAPD and ISSR (*Hamada & Hagimori, 1996*; *Zhang et al., 2009*; *Lu et al., 2012*; *Chen et al., 2013*; *Lu et al., 2014*), therefore, were used for cultivar identification and to evaluate the genetic diversity and relationship of germplasm resources of colored calla lily. However, very limited and uncertain information has been obtained until now. EST-SSR markers facilitate better cross-genome comparisons and genetic diversity and relationship evaluation because their target coding domains are more likely to be conserved between relatives (*Morgante, Hanafey & Powell, 2002*; *Varshney, Graner & Sorrells, 2005*). In the present study, 174 alleles were identified in 21 accessions using 58 polymorphic EST-SSR markers with an average of 3.0 alleles per locus. The discriminating power, as determined by the PIC value, ranged from 0.130 to 0.757 with a mean value of 0.420. *Wei et al. (2012)* reported 43 EST-SSR markers with an average gene diversity of 0.446 (*PIC*) in 24 *Z. aethiopica* individual plants. The polymorphism level here was comparable to that of the EST-SSR based study in *Z. aethiopica*.

A cluster analysis based on 58 polymorphic EST-SSR markers was then used to assess the genetic relationships among 21 accessions. Three major groups were identified at a cut-off genetic distance index of 0.18. Cluster I consisted of thirteen accessions with a complex set of colored spathes, including red, purple, and pink, among others. The cultivated species *Z. rehmannii* cv. Rehmannii and another four cultivars (Super Gem, Rose Gem, Xiangyuan Red and Allure) used in the initial validation of EST-SSR markers were grouped into several sub-clusters. These sub-clusters may reflect that *Z. rehmannii* is easily crossed with other species (*Letty, 1973*; *Funnell, 1993*; *Singh, 1996*), giving rise to hybrids with lobed leaves and spathes that vary in color and shape. Surprisingly, eight accessions with yellow

flowers were separated into Clusters II and III. The cultivars Black Magic in Cluster II and Best Gold in Cluster III are intra-species hybrids of *Z. elliottiana* and *Z. pentlandii*, respectively, representing the close relationship between Clusters II and III, as well as *Z. elliottiana* and *Z. pentlandii*. The species *Z. elliottiana* and *Z. pentlandii*, also known as Yellow Arum (*Letty, 1973*; *Singh, 1996*), have leaves that are hastate to cordate at the base and differ in their consistency of yellow spathes. These species are distant from the other species *Z. albomaculata* and *Z. rehmannii*. *Snijder, Santiago & Tuyl (2007)* determined the plastome composition of *Z. aethiopica*, *Z. rehmannii*, *Z. albomaculata* subsp. *albomaculata*, *Z. albomaculata* subsp. *macrocarpa*, *Z. elliottiana* and *Z. pentlandii* with species-specific CAPS markers developed from the plastidial intergenic region of *trn*D and *trn*C (DC). They revealed that *Z. elliottiana* and *Z. pentlandii* showed a DC-*Alu*I and a DC-*Hae*III restriction pattern that differed from that of *Z. rehmannii* and *Z. albomaculata*. The results presented here also support the above findings and suggest that EST-SSR makers developed from *de novo* transcriptome analysis are potential tools for taxonomy and cultivar identification in the genus *Zantedeschia*. Interestingly, the genetic relationships among colored calla lily accessions seem to be related to the spathe color. However, we should note that this conclusion was drawn from limited numbers of *Zantedeschia* accessions in section *Aestivae*. A higher number of accessions and more individual cultivated plants will be essential for verifying the abovementioned relationship in future studies.

## CONCLUSIONS

This study was an attempt to present the transcriptome of colored calla lily using Illumina next-generation sequencing and *de novo* assembly. A total of 39,298 unigenes with an average length of 1,038 bp were generated, of which 53.6% (21,077) were annotated using the Nr, Swiss-Prot, COG, GO and KEGG databases. Based on the transcriptome dataset, we identified a total of 117 unique transcripts related to flower development, including pigment biosynthesis, floral organ development, flowering regulation and flower senescence. Moreover, a large number of SSRs and SNPs were mined and identified, and high-quality primers of 200 SSR loci were designed and demonstrated for their amplification and cross-species transferability in germplasm resources of colored calla lily. Finally, a relatively distinct genetic relationship among 21 accessions in section *Aestivae* was elucidated via 58 EST-SSR markers. The enrichment results highlight the potential of a *de novo* transcriptome dataset for functional genomics studies and molecular marker development. The EST-SSR markers generated in this study will enhance the current repository for the genus *Zantedeschia* and will be useful for taxonomic study and crop improvement programs.

### Funding

This work was supported by National Natural Science Foundation of China (31301803), China Postdoctoral Science Foundation Project, Beijing Natural Science Foundation (6144021) and Beijing Academy of Agriculture and Forestry Sciences for Youth

(QNJJ201403). The funders had no role in study design, data collection and analysis, decision to publish, or preparation of the manuscript.

## Grant Disclosures

The following grant information was disclosed by the authors:
National Natural Science Foundation of China: 31301803.
China Postdoctoral Science Foundation Project.
Beijing Natural Science Foundation: 6144021.
Beijing Academy of Agriculture and Forestry Sciences for Youth: QNJJ201403.

## Competing Interests

The authors declare there are no competing interests.

## Author Contributions

- Zunzheng Wei conceived and designed the experiments, performed the experiments, analyzed the data, wrote the paper, reviewed drafts of the paper.
- Zhenzhen Sun performed the experiments, analyzed the data.
- Binbin Cui performed the experiments.
- Qixiang Zhang and Di Zhou conceived and designed the experiments, reviewed drafts of the paper.
- Min Xiong and Xian Wang contributed reagents/materials/analysis tools.

## DNA Deposition

The following information was supplied regarding the deposition of DNA sequences:
   Sequence Read Archive (SRA) number: SRR3310941.

## Data Availability

   The raw data has been supplied as a Supplemental Dataset.

## Supplemental Information

Supplemental information for this article can be found online at http://dx.doi.org/10.7717/peerj.2378#supplemental-information.

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
