# Peer review of "Transcriptome analysis of colored calla lily (Zantedeschia rehmannii Engl.) by Illumina sequencing: de novo assembly, annotation and EST-SSR marker development"

_PeerJ, doi:10.7717/peerj.2378_

## Round 0.1 · original submission · Major Revisions

The manuscript can be considered for publication after addressing the suggestions for improvements provided by the reviewers. Please also pay extra attention to the clarity of the English language in the manuscript.

·

Basic reporting

The article is mostly very well written and discussed. The results are greatly valuable for the molecular studies about the Zantedeschia genus, as well as to Araceae family.
I suggest some minor revisions as follows:
1. At Figure 1 legend, for the format editors, I suggest a better organization of the informations, the legend seems unformatted.
2. At Figure 2 legend, please provide more details. Each component should be described in detail. The figure should be self explained.

Experimental design

No Comments.

Validity of the findings

No Comments.

Additional comments

I believe the authors used a very safe method to obtain realiable results using RNA-Seq as deep sequencing. Once the genus studied (Zantedeschia) is still poor explored when it comes to molecular approach, the work offers precious data about non model organisms, both in the aspect of adapt methods for working with non model plants (such as the more appropriated databases to use, organisms to compare in the lack of reference genome, etc) and the results obtained itself, that provides helpful improvment for botanical and plant molecular biology reasearchers.

·

Basic reporting

The manuscript by Wei et al. (peerj-10164) describes the characterization the transcriptome and develop EST-SSR markers of colored calla lilyin this study. After sequencing a RNA library, the authors provided some evaluation of the sequencing data. Subsequently, they tried to identify and verified some potential SSR and SNP markers. Overall, the novelty of this manuscript is weak. Although the results here provide some information, but they are not significantly beyond previous findings. The manuscript suffers of several issues.

1. Though 7162 SNPs were developed as putative molecular markers. None of them were verified.

2. There are some uppercase or lowercase letters in the References section. The format of references should be coincident.

3. Line 181-183: The format is incorrect.

4. The English need improvement across the whole manuscript.

Experimental design

The dxperimental design is ressonable.

Validity of the findings

It is not significantly beyond previous findings.

Reviewer 3 ·

Basic reporting

The manuscript entitled "Transcriptome analysis of colored calla lily (Zantedeschia rehmannii Engl.) by Illumina sequencing: de novo assembly, annotation and EST-SSR marker development" by Wei et al. has provided the comprehensive transcript annotation of the flower plant and suggested genetic markers (SSR and SNP) for polymorphism studies within the genotypes.

Experimental design

No Comments

Validity of the findings

No Comments

Additional comments

1. Confirm which is correct; E-value threshold of 10-5 (line 234) in Results or E-value 1e-10-6 (line 171) in Materials and Methods.

2. In SSR analysis, please do not include mononucleotide repeats as it could be generated as homopolymers in Illumina sequencing. Please only include di- to hexa-nucleotide repeats for SSR analysis. In lines 357 & 358 they have discussed the same. Hence revise the table by deleting mononucleotide repeats.

3. The authors should explain the species distribution results summarizing 32% of top-hits to Vitis vinifera. Which database was used for species distribution analysis.

4. Under GO classification of unigenes, evidence code (EC) assignment is necessary. Each GO term is associated with a EC that mostly is inferred from computationally-derived annotation. So, in explaining GO terms, please discuss the results as suggested functions. Please discuss the GO results under this perspective.

5. Line No-308. How did the authors recover unigenes related to flower development based on literature search.

6. The authors should provide the functional annotation of SSR containing sequences of lengths more than 1 kb as a separate Table to supplementary information. Then discuss on the transferability of SSRs to other related species.

---

## Round 0.2 · accepted · Accept

Congratulations for acceptance of your manuscript. There is still one typing error to be corrected in the file for production:

line 537: 'predicated' should be 'predicted'

·

Basic reporting

No Comments

Experimental design

No Comments

Validity of the findings

As I previously pointed, I believe the authors used a very safe method to obtain realiable results using RNA-Seq as deep sequencing. the results are interesting mainly because this genus (Zantedeschia) is very neglected at the literature. Even if the approach is today very well explored, the study main focus still poor explored. These results will provide helpful a great acquisition to plant molecular biology and biochemistry reasearchers.

·

Basic reporting

No Comments

Experimental design

No Comments

Validity of the findings

No Comments

Additional comments

The manuscript by Wei et al. (peerj-10164) describes the characterization the transcriptome and develop EST-SSR markers of colored calla lilyin this study. After sequencing a RNA library, the authors provided some evaluation of the sequencing data. They tried to identify and verified some potential SSR and SNP markers. It is acceptable.